EMBO
Molecular Medicine

# Comparative proteomics reveals a diagnostic signature for pulmonary head-and-neck cancer metastasis

Hanibal Bohnenberger[1,†], Lars Kaderali[2,†], Philipp Ströbel[1], Diego Yepes[3,4], Uwe Plessmann[5], Neekesh V Dharia[6,7,8] (iD), Sha Yao[1], Carina Heydt[9], Sabine Merkelbach-Bruse[9], Alexander Emmert[10], Jonatan Hoffmann[1], Julius Bodemeyer[1], Kirsten Reuter-Jessen[1], Anna-Maria Lois[1], Leif Hendrik Dröge[11], Philipp Baumeister[12], Christoph Walz[13], Lorenz Biggemann[14], Roland Walter[3], Björn Häupl[3,4], Federico Comoglio[15,16], Kuan-Ting Pan[5], Sebastian Scheich[3], Christof Lenz[5,17], Stefan Küffer[1], Felix Bremmer[1], Julia Kitz[1], Maren Sitte[18], Tim Beißbarth[18], Marc Hinterthaner[10], Martin Sebastian[3], Joachim Lotz[14,19], Hans-Ulrich Schildhaus[1], Hendrik Wolff[20,21], Bernhard C Danner[10], Christian Brandts[3,4], Reinhard Büttner[9], Martin Canis[12], Kimberly Stegmaier[6,7,8], Hubert Serve[3,4], Henning Urlaub[5,14] & Thomas Oellerich[3,4,*] (iD)

## Abstract

Patients with head-and-neck cancer can develop both lung metastasis and primary lung cancer during the course of their disease. Despite the clinical importance of discrimination, reliable diagnostic biomarkers are still lacking. Here, we have characterised a cohort of squamous cell lung (SQCLC) and head-and-neck (HNSCC) carcinomas by quantitative proteomics. In a training cohort, we quantified 4,957 proteins in 44 SQCLC and 30 HNSCC tumours. A total of 518 proteins were found to be differentially expressed between SQCLC and HNSCC, and some of these were identified as genetic dependencies in either of the two tumour types. Using supervised machine learning, we inferred a proteomic signature for the classification of squamous cell carcinomas as either SQCLC or HNSCC, with diagnostic accuracies of 90.5% and 86.8% in cross- and independent validations, respectively. Furthermore, application of this signature to a cohort of pulmonary squamous cell carcinomas of unknown origin leads to a significant prognostic separation. This study not only provides a diagnostic proteomic signature for classification of secondary lung tumours in HNSCC patients, but also represents a proteomic resource for HNSCC and SQCLC.

1  Institute of Pathology, University Medical Center, Göttingen, Germany
2  Institute of Bioinformatics, University Medicine Greifswald, Greifswald, Germany
3  Department of Medicine II, Hematology/Oncology, Goethe University, Frankfurt, Germany
4  German Cancer Research Center, German Cancer Consortium, Heidelberg, Germany
5  Bioanalytical Mass Spectrometry Group, Max Planck Institute for Biophysical Chemistry, Göttingen, Germany
6  Department of Pediatric Oncology, Dana-Farber Cancer Institute, Boston, MA, USA
7  Division of Pediatric Hematology/Oncology, Boston Childrens Hospital, Boston, MA, USA
8  The Broad Institute, Cambridge, MA, USA
9  Institute of Pathology, University Hospital Cologne, University of Cologne, Köln, Germany
10  Department of Thoracic and Cardiovascular Surgery, University Medical Center, Göttingen, Germany
11  Department of Radiooncology, University Medical Center, Göttingen, Germany
12  Department of Otorhinolaryngology, University Hospital Munich, Ludwig-Maximilian-University München, München, Germany
13  Institute of Pathology, Faculty of Medicine, LMU Munich, Munich, Germany
14  Institute for Diagnostic and Interventional Radiology, University Medical Center, Göttingen, Germany
15  Department of Haematology, University of Cambridge, Cambridge, UK
16  Cambridge Institute for Medical Research, Wellcome Trust/MRC Stem Cell Institute, Cambridge, UK
17  Bioanalytics, University Medical Center, Göttingen, Germany
18  Institute of Medical Statistics, University Medical Center, Göttingen, Germany
19  German Cardiovascular Research Center, Deutsches Zentrum für Herz-Kreislaufforschung (DZHK), Partnersite Göttingen, Germany
20  University Medical Center, Göttingen, Germany
21  Department of Radiooncology, University Medical Center, Regensburg, Germany
   *Corresponding author. Tel: +496963016148; E-mail: thomas.oellerich@kgu.de
   †These authors contributed equally to this work as first authors

**Keywords**  Biomarker; head-and-neck cancer; lung cancer; metastasis; proteomics

**Subject Categories**  Cancer; Post-translational Modifications, Proteolysis & Proteomics

## Introduction

Head-and-neck squamous cell carcinoma (HNSCC) affects more than 500,000 patients every year worldwide (Ferlay *et al*, 2010; Jemal *et al*, 2011; Chaturvedi *et al*, 2013). Besides local recurrences, long-term survival of patients with resectable HNSCC is limited by frequently occurring distant metastases, of which up to 80% occur in the lung (Calhoun *et al*, 1994; Jones *et al*, 1995; de Bree *et al*, 2000; Ferlito *et al*, 2001; Liao *et al*, 2007). Thus, patients with HNSCC are screened for lung metastasis after tumour resection (de Bree *et al*, 2000; Merkx *et al*, 2002). However, because tobacco-smoking is not only a major risk factor for the development of HNSCC, but also for lung cancer, HNSCC patients are furthermore at high risk of developing metachronous squamous cell carcinomas of the lung (SQCLC; Talamini *et al*, 2002). Differentiation between lung metastasis of HNSCC (metHNSCC) and SQCLC is of clinical importance, as the diagnosis guides the therapeutic procedures that can range from curative treatment for SQCLC patients to palliative treatment for patients with metastatic HNSCC (Atabek *et al*, 1987; Jacobs *et al*, 1992; Jones *et al*, 1995; Kuriakose *et al*, 2002; Henschke *et al*, 2003; Battafarano *et al*, 2004; Wisnivesky *et al*, 2004; Pignon *et al*, 2008; Shiono *et al*, 2009). Despite this clinical importance, reliable biomarkers for differentiation between SQCLC and HNSCC metastasis in the lung are currently lacking; this may be due, at least partly, to the overlapping aetiology, morphology and biology of these tumour entities. Because decision-making by clinicians currently relies on non-validated clinical and imaging criteria, there is an urgent need for reliable molecular biomarkers that can differentiate between SQCLC and HNSCC lung metastases (Geurts *et al*, 2005).

Head-and-neck squamous cell carcinoma and SQCLC have been extensively studied at the genomic and transcriptomic levels. However, in addition to their shared morphology, HNSCC and SQCLC exhibit largely overlapping patterns of genetic mutations and copy number alterations (van Oijen *et al*, 2000; Tabor *et al*, 2002; Geurts *et al*, 2005, 2009; Talbot *et al*, 2005; Vachani *et al*, 2007; Cancer Genome Atlas Research, 2012; Cancer Genome Atlas, 2015; Ichinose *et al*, 2016a). Given the apparent inability of genomic features to differentiate reliably between HNSCC and SQCLC, we have investigated a clinically and genetically well-characterised cohort of SQCLC, HNSCC and undetermined lung tumours by quantitative mass spectrometry-based proteomics. We reasoned that this proteomic approach, rather than genomic and transcriptomic studies, may identify a suitable biomarker panel and may also provide some more general insights into the biology of these tumours.

## Results

### Comparative proteomic characterisation of squamous cell carcinomas of the lung and the head-and-neck region

Differentiation between metHNSCC and primary SQCLC is of fundamental clinical importance for therapeutic stratification. However, diagnostic biomarkers are so far lacking, owing to the large number of morphological and genomic features that are shared by these tumour entities.

Because the two tumour types have not been systematically compared at the proteome level, we wished to explore the possibility of identifying proteomic diagnostic biomarkers. To this end, we characterised the protein expression profiles of 44 formalin-fixed and paraffin-embedded (FFPE) SQCLC tissues and 30 FFPE HNSCC tissues from patients who developed squamous cell tumours in the lung in the course of their disease. All SQCLCs in our cohort were in tumour stage I–III with grades 2–3, and the tumour stage of HNSCCs ranged from I to IV with grades 2–3 (see Table 1 and Dataset EV1 for detailed patient characteristics). All patients were treated by surgery and 40 of them had received additional adjuvant (chemo)radiotherapy. None of the patients had received neoadjuvant therapy or primary chemoradiotherapy. Squamous cell histology of all samples was confirmed by expert pathology review including immunohistochemical staining of the markers p63 and cytokeratin 5/6 (Fig 1A). Moreover, targeted next-generation sequencing revealed similar mutation patterns in both HNSCC and SQCLC samples from our cohort (Fig 1B). This is in accordance with genomic studies that have reported similar somatic mutation patterns and frequencies in both diseases, as illustrated in Fig 1C (Cancer Genome Atlas Research N, 2012; Cancer Genome Atlas, 2015). The mutations detected affected mostly the PI3-kinase (PI3K) and Ras pathways, receptor tyrosine kinases (RTK), TP53 and the NFE2L2/KEAP1 pathway (Fig 1B and Dataset EV1). Notably, three HNSCC samples were positive for human papillomavirus (HPV) 16.

For the proteomic characterisation of the 74 tumour samples (44 SQCLC, 30 HNSCC), we combined a filter-aided sample preparation approach (FASP) with Super-SILAC-based quantitative mass spectrometry (Fig 2A; Wisniewski *et al*, 2009; Erde *et al*, 2014; Bohnenberger *et al*, 2015). Before protein extraction, tumour cell areas were marked under microscopic view by expert pathologists, and the tumour cell content was enriched to >80% by macrodissection, which is a well-established technique for the molecular analysis of tumour tissue samples in routine pathology diagnostics (Cree *et al*, 2014). After FASP-based protein extraction, the tumour-derived proteins were mixed in equimolar amounts with a Super-SILAC spike-in protein standard that was used for relative quantification of protein abundance. The Super-SILAC standard consisted of the four lung cancer cell lines NCI-H2228, HCC15, HCC44 and NCI-H1339. It represented adequately the proteomes of SQCLC and HNSCC samples, as reflected by the fact that more than 90% of the SILAC ratios were within a fivefold range, allowing accurate and comparable protein quantification (Fig 2B). The protein mixture was digested with the protease trypsin, and the resulting peptides were finally analysed by high-resolution mass spectrometry.

We quantified a total of 6214 proteins in SQCLC and HNSCC samples (Datasets EV2 and EV6), thus providing the largest proteome resource for squamous cell carcinomas to date. On

**Table 1. Patient characteristics training cohort.**

| Characteristic | Squamous cell lung carcinoma (*n* = 44) | Squamous cell head-and-neck carcinoma (*n* = 30) |
|---|---|---|
| Age median (range) [years] | 66 (49–81) | 55 (31–76) |
| Male sex [no. (%)] | 32 (72.7) | 26 (86.7) |
| Smoking or tobacco use [no. (%)] | | |
| Current or former | 22 (50.0) | 28 (93.3) |
| Never | 0 (0.0) | 2 (6.7) |
| Not reported | 22 (50.0) | 0 (0.0) |
| Site of primary tumour [no. (%)] | | |
| Lung | 44 (100.0) | — |
| Larynx | — | 12 (40.0) |
| Oral cavity | — | 5 (16.7) |
| Pharynx | — | 13 (43.4) |
| Systemic therapy regimen [no. (%)] | | |
| Adjuvant therapy | 22 (50.0) | 18 (60.0) |
| Neoadjuvant therapy | 0 (0.0) | 0 (0.0) |
| Overall survival: median follow-up time (range) [months] | 19 (1–38) | 57 (7–169) |
| No. of reported deaths (%) | 17 (31.8) | 20 (66.7) |
| Local relapse of HNSCC [no. (%)] | — | 5 (16.7) |
| Time HNSCC until lung tumour: Median (range) [months] | — | 20 (1–121) |
| Staging UICC (7th edition) | | |
| Stage I [no. (%)] | 14 (31.8) | 1 (3.3) |
| Stage II [no. (%)] | 11 (25.0) | 5 (16.7) |
| Stage III [no. (%)] | 19 (43.2) | 7 (23.3) |
| Stage IV [no. (%)] | 0 (0.0) | 17 (56.7) |
| TP53 sequence | | |
| Wild type [no. (%)] | 12 (27.3) | 12 (40.0) |
| Mutated [no. (%)] | 31 (70.5) | 12 (40.0) |
| Unknown | 1 (2.2) | 6 (20.0) |

average, we quantified 2,251 (range 1,317–3,314) proteins in a total of two replicates of each of the tissue samples analysed (Fig 2C). Notably, the correlation coefficients for the individual replicates ranged from 0.802 to 0.975 (0.949 ± 0.0266; Fig EV1A), and the MS signal intensities spanned six orders of magnitude, which reflects the identification and quantification of proteins with high and those with low abundance (Fig 2D). Moreover, the results obtained on different mass spectrometry platforms were comparable and reproducible, as revealed by a direct comparison of the protein expression profiles of six randomly selected tumour samples that were analysed on both an Orbitrap Fusion and a Q Exactive HF mass spectrometer (both Thermo Fisher; Fig EV1B and C).

As a first exploratory analysis, we performed a principal component analysis on the full proteomic data. This revealed significant proteomic differences between HNSCC and SQCLC, as it separated the two tumour types into two partially overlapping groups reflecting the cell of origin of the respective squamous cell carcinomas (Fig 2E).

**Quantitative proteomics reveals differences in protein expression between SQCLC and HNSCC**

In our proteomic dataset, we identified 518 proteins with significantly different expression levels in HNSCC and SQCLC by univariate data analysis using an empirical Bayes method with a linear regression model (Figs 3A and EV2A, and EV3A, Dataset EV3). In line with previously published gene expression and immunohistochemistry data, cytokeratin 19 (CK19) was among the most strongly up-regulated proteins in our SQCLC cohort (Ichinose *et al*, 2016a; Figs 3A and EV2B). Interestingly, previously described oncogenic factors—such as HMG-CoA synthase 1 (HMGCS-1) and FAM83H—were also found to be differentially expressed in SQCLC and HNSCC. These proteins have been described as being overexpressed and relevant for tumour cell survival in various cancer entities (Snijders *et al*, 2017; Zhao *et al*, 2017). To systematically investigate the functional relevance of the proteins that were found to be differentially expressed in SQCLC and HNSCC, we have integrated our proteomic

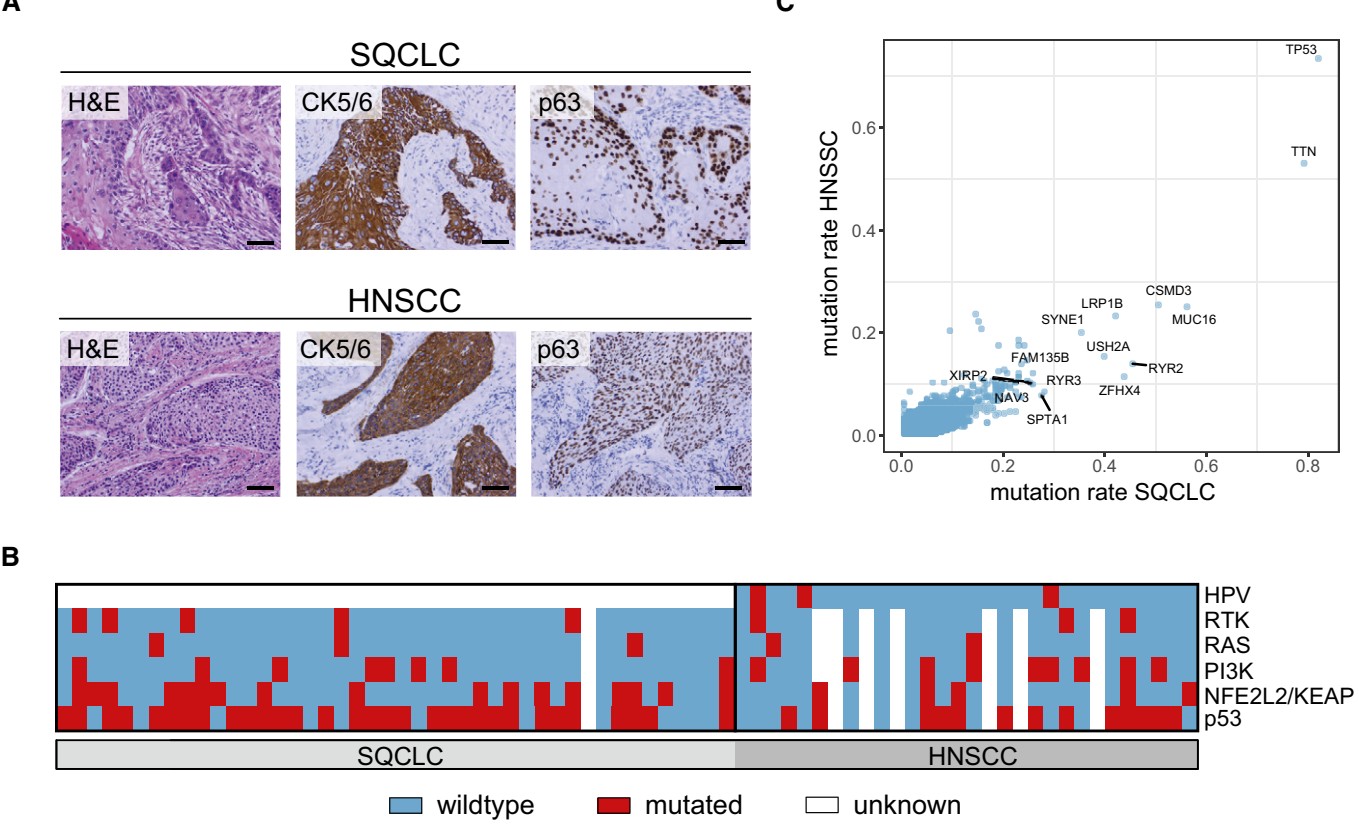

**Figure 1. Genetic comparison of lung and head-and-neck carcinomas.**

A   Representative H&E and immunohistochemical stainings of CK5/6 and p63 in pulmonary and head-and-neck squamous cell carcinomas. Scale bar: 500 μm.

B   Heatmap showing somatic mutations in defined pathways and HPV status for all HNSCC and SQCLC cases analysed. (Blue, no mutation/no HPV detected; red, mutation/HPV detected; white, no data available)

C   Comparison of mutation rates in HNSCC (n = 279) and SQCLC (n = 178) cases from TCGA. Genes exhibiting a mutation rate > 0.25 in both cancer types are labelled.

results with recently published functional genomic data from genome-scale CRISPR/Cas9-based loss-of-function screens (Doench et al, 2016; Meyers et al, 2017; https://figshare.com/articles/Broad_Institute_Cancer_Dependency_Map_CRISPR_Avana_dataset_18Q1_Avana_public_18Q1_/5863776). To this end, we have investigated if any of the 518 differentially expressed proteins were identified as genetic dependencies (i.e. regulators of proliferation and/or cell survival) in 12 HNSCC and 10 SQCLC cell lines (for details, see Dataset EV3). Interestingly, this analysis revealed that a subset of the differentially expressed genes/proteins shows preferential genetic dependencies in either HNSCC or SQCLC (Fig 3B, Dataset EV3). For instance, the mitotic checkpoint protein BUB3 and dolichol-phosphate mannosyltransferase subunit (DPM1) were stronger dependencies in HNSCC, while the adenylosuccinate lyase (ADSL), the cytochrome b-c1 complex subunit 1 (UQCRC1) and the NADH dehydrogenase 1 beta subcomplex subunit 4 (NDUFB4) belonged to a subset of proteins that reflected stronger dependencies in SQCLC. Notably, a genetic dependency on some of the most differentially expressed proteins was observed (Fig EV3B). This result highlights the tumour-type-specific functional relevance of a subset of the differentially expressed marker proteins.

Next, we validated our proteomic data by performing immuno-histochemical staining for CK19, HMGCS-1, FAM38H, LGALS7 and ferritin light chain (FTL) in 212 SQCLC and 343 HNSCC tissues (Table 2 and Dataset EV4), which confirmed our proteomic data in that CK19, HMGCS-1 and FTL were more strongly expressed in SQCLC, as opposed to LGALS7 and FAM83H, both of which showed stronger expression in HNSCC (Figs 3C and D, and EV2B). To exclude the possibility that the observed differential expression of FTL was caused by differences in microvessel densities between SQCLC and HNSCC, we performed CD34 staining, which confirmed similar microvessel densities in the two tumour types (Fig EV2C). Notably, no differences in expression of proteins related to the most recurrently mutated pathways shown in Fig 1B and C were observed between SQCLC and HNSCC (Fig EV2D).

On the basis of the described results, we performed a gene-set-enrichment analysis of the 518 differentially expressed proteins to gain insight into pathways whose regulation differs between SQCLC and HNSCC. This revealed that (i) vesicle transport, (ii) glycosylation and (iii) RNA-processing including splicing (each adjusted P value was < 0.001) were among the most differentially regulated pathways in SQCLC and HNSCC (Fig 3E). Our proteomic data suggest that these processes are regulated differentially in SQCLC and HNSCC cells and might contribute to their pathophysiology.

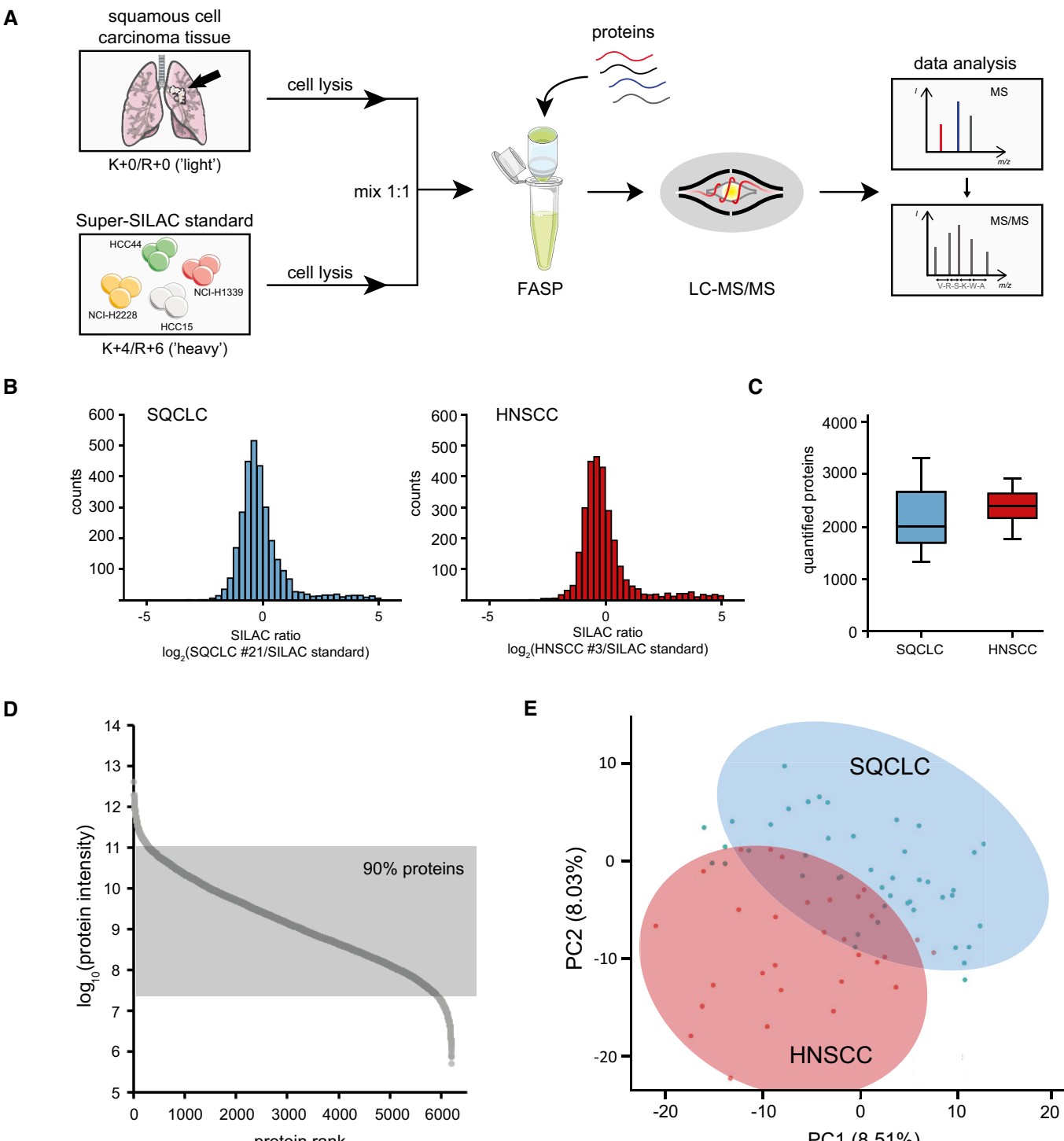

**Figure 2.  Proteomic comparison of lung and head-and-neck carcinomas.**

A   Schematic experimental workflow: Patient-derived tumour samples were lysed according to the FASP protocol; the resulting proteins were mixed in equimolar amounts with a Super-SILAC spike-in standard and digested with trypsin. The MS analysis of each sample was performed in duplicate, and data analysis was performed with the software packages MaxQuant, Perseus and R.

B   Representative histogram showing the SILAC ratio distribution between the Super-SILAC standard and SQCLC (left) and HNSCC (right) samples.

C   Boxplot showing the numbers of quantified proteins, derived from 44 SQCLC and 30 HNSCC tissue samples. The central line in the boxes represents the median number of proteins over all samples, and upper and lower borders of boxes correspond to 25% and 75% quantiles. Whiskers indicate minimum and maximum.

D   Distribution of MS-based protein signal intensities in all 74 tissue samples of the training cohort.

E   Principal component analysis of SQCLC (n = 44; blue) and HNSCC (n = 30; red) protein expression. Shown are the first two principal components, accounting for 8.51% and 8.03% of the total variance in the data, respectively.

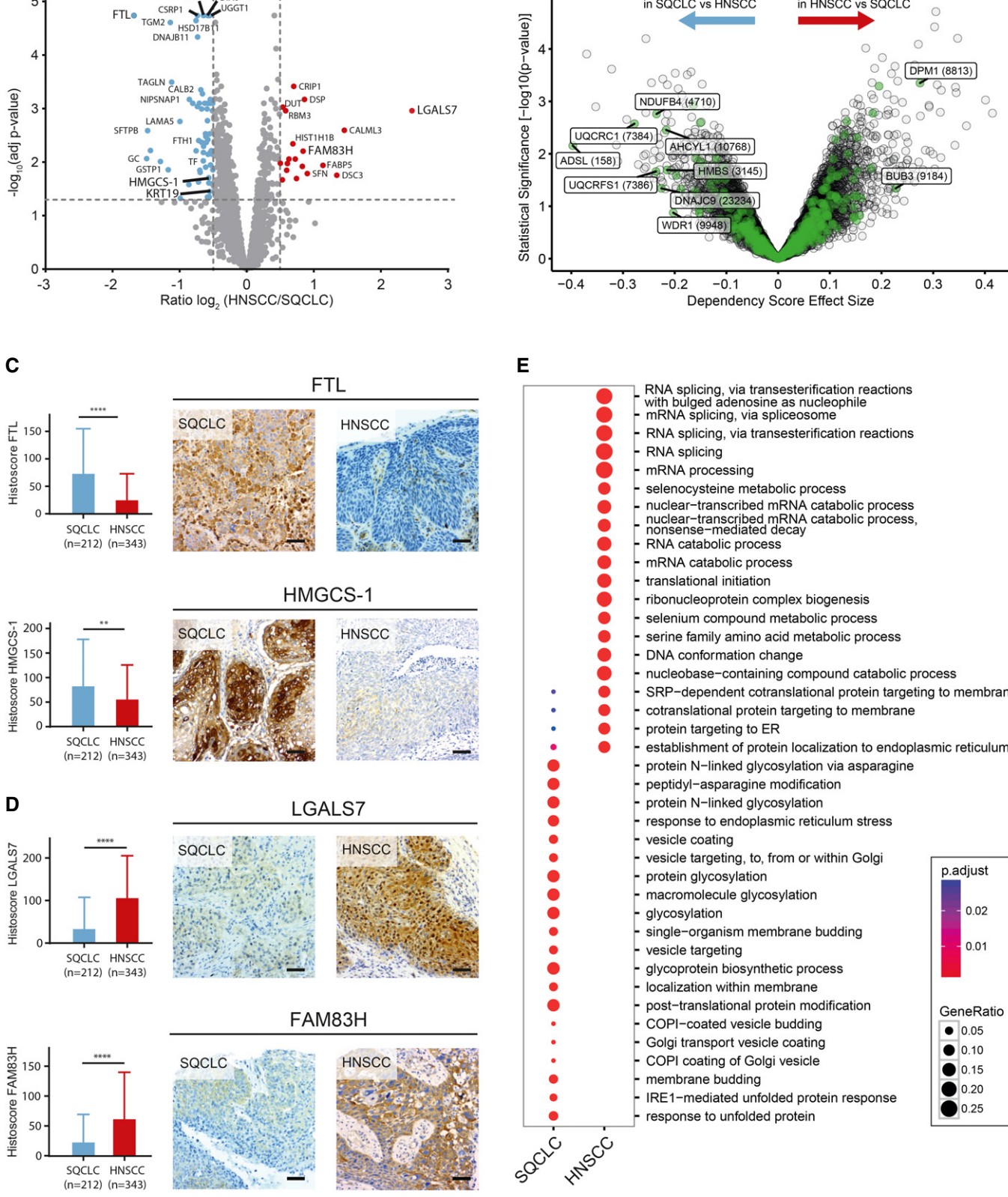

**Figure 3.**

**Figure 3.   Validation of proteomic differences of lung and head-and-neck carcinomas.**

A    Volcano plot relating adjusted *p* values for differential protein expression to averaged normalised SILAC ratios of two replicates. Red (higher expression in HNSCC) and blue (higher expression in SQCLC) dots indicate significantly regulated proteins (*P* < 0.05).

B    Two-class comparison of genetic dependencies from a publically available genome-scale CRISPR-Cas9 screen of HNSCC (12 cell line) versus SQCLC (10 cell lines) identified a subset of differentially expressed proteins that were also differential dependencies in these tumour types. The *x*-axis represents the effect size of the mean difference of dependency scores in HNSCC compared to SQCLC cell lines. Positive effect size indicates a greater mean dependency in HNSCC; negative effect size indicates a greater mean dependency in SQCLC. The *y*-axis represents the statistical significance of differential enrichment calculated as -log10(*P*-value) from a two-sided *t*-test. The *P*-values used for this plot are uncorrected for multiple hypothesis testing. Highlighted in green are genetic dependencies that were also identified as differentially expressed proteins in our study.

C, D    Immunohistochemical analysis of the expression of HMGCS-1 (*P* = 0.0014), FTL (*P* = 0.0001) (C), LGALS7 (*P* = 0.0001) and FAM83H (*P* = 0.0001) (D) in an independent cohort of 212 SQCLC and 343 HNSCC cases. Scale bar indicates 100 μm. Shown are mean values and standard deviation. Statistical significance was assessed using Wilcoxon–Mann–Whitney test.

E    Pathway enrichment analysis for proteins differentially expressed in HNSCC and SQCLC.

**Table 2.   Patient characteristics immunohistochemistry cohort.**

| Characteristic | Squamous cell lung carcinoma (*n* = 212) | Squamous cell head-and-neck carcinoma (*n* = 343) |
|---|---|---|
| Age median (range) [years] | 66 (42–83) | 57 (20–88) |
| Male sex [no. (%)] | 178 (83.9) | 277 (80.8) |
| Site of primary tumour [no. (%)] | | |
| Lung | 212 (100.0) | — |
| Larynx | — | 80 (23.3) |
| Oral cavity | — | 107 (31.2) |
| Pharynx | — | 156 (45.5) |
| Staging UICC (7th edition) | | |
| Stage I [no. (%)] | 88 (41.5) | 14 (4.1) |
| Stage II [no. (%)] | 63 (29.7) | 33 (9.6) |
| Stage III [no. (%)] | 59 (27.8) | 49 (14.3) |
| Stage IV [no. (%)] | 2 (0.9) | 196 (57.1) |

## A diagnostic proteomic signature for differentiation between primary SQCLC and head-and-neck cancer metastasis in the lung

As we were aiming to obtain a robust diagnostic signature with high classification accuracy, we next trained a linear support vector machine (SVM) to discriminate between SQCLC and HNSCC (Fig 4A). Training was performed using a bootstrapping procedure to assess robustness of predictions and guard against overfitting. Using the full set of proteins as features in a crossvalidation approach, the predictor achieved a bootstrapped accuracy of 90.5 ± 0.05, a sensitivity of 85.7% and a specificity of 93.7%. For comparison, we used four other classification algorithms, which achieved comparable performance (Fig 4B). We then assessed the extent to which reducing the size of the signature set would affect classification performance. Using recursive feature elimination, we were able to reduce the set to about 1,100 proteins without affecting classification performance. Thereafter, the accuracy of the classification began to decline gradually, with a bootstrapped accuracy of 76.8% using 100 proteins, 65% using the top 25 proteins and only 57.2% using a single protein for classification (Fig 4C). This indicates clearly that reliable distinction

**Figure 4.   Diagnostic proteomic signature for HNSCC metastasis in the lung.**

A    Schematic computational workflow: The SVM predictor was initially developed using a bootstrapping procedure on *n* = 44 primary SQCLC and *n* = 30 HNSCC cases. An independent set of *n* = 38 tumours was used for validation, before we applied the classifier to *n* = 51 lung tumours of unknown origin. Survival analysis was used to assess classification performance on this final dataset.

B    Classification accuracy of five different machine-learning methods used for classification of HNSCC vs. SQCLC is as follows: SVM, support vector machine with a linear kernel; RPART, classification and regression tree (CART); NB, naïve Bayes classifier; PAM, partitioning around medoids/nearest shrunken centroids; and PLR, L1-penalised logistic regression. All models were fitted using the R/Bioconductor package "caret". Model hyperparameters were optimised using caret's tuning option; accuracies were computed by bootstrapping with 25 repetitions, shown are mean accuracy ± standard deviation.

C    Classification accuracy using a linear support vector machine. The figure shows the bootstrapped accuracy achieved over the number of proteins included in the computation. Feature selection was performed using recursive feature elimination.

D    Boxplot showing the number of quantified proteins derived from 19 SQCLC, 19 HNSCC and 51 squamous cell lung tumours of unknown origin derived from HNSCC patients. The central line in the boxes represents the median number of proteins over all samples, and upper and lower borders of boxes correspond to 25% and 75% quantiles. Whiskers indicate minimum and maximum.

E    Classification accuracy of the SVM prediction model described, in an independent validation cohort of 19 SQCLC and 19 HNSCC cases.

F    Representative H&E and immunohistochemical stainings of CK5/6 and p63 in squamous cell lung tumours of unknown origin derived from HNSCC patients. Scale bar: 500 μm.

G    Kaplan–Meier analysis of overall survival (OS) in which all patients with lung tumours were grouped according to the clinical classification (left, median survival SQCLC: 55 months; metHNSCC: 31 months) or proteomic classification (right, median survival SQCLC: 55 months; metHNSCC: 8 months) of the tumour as metHNSCC or SQCLC. The *P* value is from a log-rank test.

H    Kaplan–Meier analysis of overall survival (OS) in which only the patients that were independent from the training cohort were grouped according to the clinical classification (left, median survival SQCLC: 84 months; metHNSCC: 35 months) or proteomic classification (right, median survival SQCLC: 84 months; metHNSCC: 5 months) of the tumour as metHNSCC or SQCLC. The *P* value is from a log-rank test.

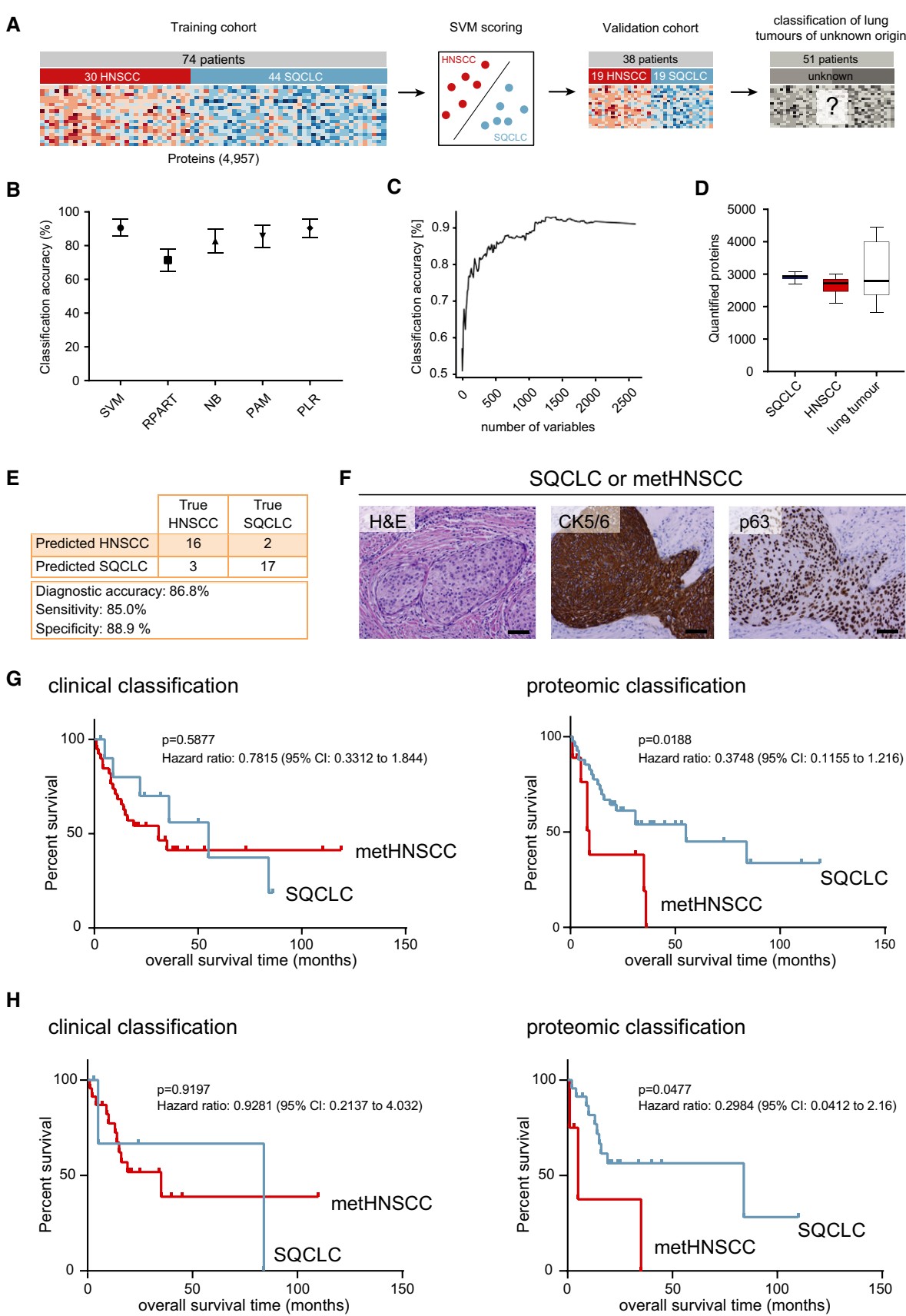

Figure 4.

between these closely related tumour types will only be possible if complex protein expression signatures, rather than single immunohistochemical markers, are used.

In the next step, and as an independent validation of our diagnostic proteomic signature, we analysed the protein expression profiles of 38 independent tumour samples, of which 19 were either confirmed primary SQCLC or primary HNSCC cases. A median of 2841 (range 2,111–3,067) proteins was quantified in each of the 38 samples (Fig 4D), and we subsequently applied our diagnostic signature to classify the tumours as either SQCLC or HNSCC on the basis of their protein expression profiles. Similar to the crossvalidation analysis, the diagnostic accuracy in this independent validation cohort was 86.8%, the sensitivity 85% and the specificity 88.9% (Fig 4E).

On the basis of these results, we finally applied our proteomic approach to classify 51 squamous cell lung tumours of undetermined origin derived from patients with a history of HNSCC as either primary SQCLC or metHNSCC. Again, squamous cell histology of all samples was confirmed by histomorphology and immunohistochemical staining (Fig 4F), and targeted next-generation sequencing revealed mutation patterns similar to those for HNSCC and SQCLC (Fig EV4A). We quantified a median of 2787 proteins

(range 1,830–4,444) in each of the tissue samples (Fig 4D) and, again, more than 90% of the SILAC ratios were within a fivefold range (Fig EV4B).

It should be noted that 23 of these secondary tumours were derived from patients whose primary HNSCC tumours were included in the training cohort of HNSCC patients described above. Of all the tumours analysed, 40 had been clinically classified as metHNSCC and 11 as primary lung cancer according to criteria of a classification score that is used in clinical routine (Fig EV4C; Ichinose *et al*, 2016b). This score is based on various clinical variables, including local relapse of the HNSCC, the time between HNSCC and the lung tumour and the number of tumour sites in the lung (see Table 3 and Dataset EV5 for details of the clinical samples). As a surrogate marker, a prognostic disadvantage would be expected for patients with metHNSCC compared with localised primary SQCLC. However, application of the clinical classifier did not show any differences in survival (Fig 4G, left). In contrast, when the SVM classifier was used to assign the samples to either HNSCC or SQCLC, significant differences in overall survival were observed between the two groups, with median survival of 8 months in the predicted metHNSCC patients versus a median survival of 55 months in the predicted SQCLC

**Table 3.  Patient characteristics validation cohorts.**

| Characteristic | Squamous cell lung carcinoma (*n* = 19) | Squamous cell head-and-neck carcinoma (*n* = 19) | Lung tumour of unknown origin after HNSCC (*n* = 51) |
|---|---|---|---|
| Age median (range) [years] | 64 (43–80) | 58 (44–78) | 63 (31–78) |
| Male sex [no. (%)] | 16 (84.2) | 18 (94.7) | 43 (84.3) |
| Smoking or tobacco use [no. (%)] | | | |
| Current or former | 19 (100.0) | 15 (78.9) | 41 (80.4) |
| Never | 0 (0.0) | 4 (21.1) | 3 (5.9) |
| Not reported | 0 (0.0) | 0 (0.0) | 7 (13.7) |
| Site of primary tumour [no. (%)] | | | |
| Lung | 19 (100.0) | — | — |
| Larynx | — | 6 (31.6) | 13 (25.5) |
| Oral cavity | — | 4 (21.1) | 10 (19.6) |
| Pharynx | — | 9 (47.3) | 28 (54.9) |
| Systemic therapy regimen [no. (%)] | | | |
| Adjuvant therapy | 9 (47.4) | 19 (100.0) | 24 (47.0) |
| Neoadjuvant therapy | 0 (0.0) | 0 (0.0) | 0 (0.0) |
| Overall survival: median follow-up time (range) [months] | 48 (8–194) | 23 (8–45) | 20 (1–119) |
| No. of reported deaths (%) | 12 (63.2) | 3 (15.8) | 26 (50.9) |
| Local relapse of HNSCC [no. (%)] | — | — | 8 (15.7) |
| Time HNSCC until lung tumour: Median (range) [months] | — | — | 17 (0–139) |
| Staging UICC (7th edition) | | | |
| Stage I [no. (%)] | 10 (52.6) | 0 (0.0) | 22 (43.1) |
| Stage II [no. (%)] | 3 (15.8) | 2 (10.5) | 5 (9.8) |
| Stage III [no. (%)] | 6 (31.6) | 7 (36.8) | 17 (33.3) |
| Stage IV [no. (%)] | 0 (0.0) | 10 (52.6) | 0 (0.0) |

patients (Fig 4G, right, log-rank $P$ = 0.0188, hazard ratio = 0.3748, 95% CI: 0.1155–1.216). Notably, one lung tumour was HPV16-positive and was identified both clinically and by proteomic classification as metHNSCC. Furthermore, when we excluded all cases that were related to the training cohort, we observed a similar prognostic separation using the proteomic classifier only (Fig 4H).

In summary, our proteomic analysis revealed a robust multiparametric diagnostic signature consisting of more than 1,000 proteins for differentiation between primary SQCLC and metHNSCC.

## Discussion

Patients with squamous cell carcinomas of the head and neck can develop metachronous lung metastases (metHNSCC), but also independent primary squamous cell lung cancers (SQCLC). Differentiation between these two conditions is of fundamental clinical importance for therapeutic stratification. However, owing to the large number of morphological and genomic features that are shared by these tumour entities, diagnostic biomarkers are lacking so far and the clinical criteria currently used for differentiation between these tumour types are poorly validated and unreliable (Ichinose *et al*, 2016b).

Earlier studies that analysed loss of heterozygosity, p53 mutation status, gene expression, copy number alterations or protein expression by immunohistochemistry (Geurts *et al*, 2005, 2009; Talbot *et al*, 2005; Ichinose *et al*, 2016a; Campbell *et al*, 2018) likewise failed to distinguish reliably between metHNSCC and SQCLC. Therefore, we here used a Super-SILAC-based quantitative mass spectrometry approach, to compare the protein expression profiles of formalin-fixed and paraffin-embedded (FFPE) SQCLC and HNSCC tissues, with the aim of establishing a diagnostic biomarker signature. In smaller patient cohorts of less than 45 patients, a similar approach has been successfully used to reveal proteomic diagnostic signatures that distinguish between breast cancer and lymphoma subtypes (Deeb *et al*, 2015; Tyanova *et al*, 2016).

Our comparative proteomic approach revealed 518 proteins with significantly different expression levels in HNSCC and SQCLC. We were able to validate the differential expression of selected proteins including CK19, FAM83H, HMGCS-1, LGALS7 and FTL by immunohistochemistry, demonstrating the discovery potential of the mass spectrometry-based approach. Interestingly, HMGCS-1 and FAM83H have previously been described as oncogenic factors. FAM83H expression was shown to be up-regulated in lung-, breast-, liver- and many more cancer types, and its expression was correlated with a poor prognosis in head-and-neck cancer (Snijders *et al*, 2017; Zhao *et al*, 2017). HMGCS-1 activity was shown to be important for oncogenic survival-promoting signalling by mutant B-Raf, highlighting the direct involvement of HMGCS-1 in oncogenic signalling pathways (Snijders *et al*, 2017; Zhao *et al*, 2017). Interestingly, intersection of our proteomic data with functional genomic data revealed that some of the differentially expressed marker proteins were genetic dependencies in either of the two tumour types. Furthermore, gene-set-enrichment analysis of the 518 differentially expressed proteins revealed RNA-processing including splicing as one of the pathways showing the greatest difference in regulation

between SQCLC and HNSCC. In this context, we note that altered phosphoproteomic profiles affecting splicing factors have previously been shown to be functionally relevant for the survival of HNSCC cancer cell lines (Radhakrishnan *et al*, 2016).

Despite these marked and reproducible differences, the distinction between SQCLC and metHNSCC is not trivial. Our data show that a distinction with clinically actionable accuracy requires a diagnostic signature rather than single markers. By training a linear support vector machine to discriminate between SQCLC and HNSCC, we achieved a diagnostic bootstrapped accuracy of 90.5% on the training data and 86.8% in an independent validation cohort. The minimum number of features required to achieve this accuracy was 1,100 proteins. Importantly, in contrast to all established clinical criteria used for this purpose, our proteomic signature was able to stratify patients with lung tumours of doubtful origin into risk groups with significantly different prognosis. In detail, patients with tumours that our signature had classified as "probably metastatic HNSCC" had a poorer outcome than patients with tumours that had been classified as "probably primary SQCLC"; this is in good accordance with the predicted clinical course of the two conditions.

This finding underscores the assertion that a reliable distinction between such closely related tumour types is possible if multiparametric protein expression signatures are used. Similar observations have previously been made for other closely related tumour types, for instance diffuse large B-cell lymphomas, which can be differentiated as either activated B-cell-like or germinal centre B-cell-like lymphomas only by complex gene expression signatures, while less complex diagnostic markers appear to be unreliable (Alizadeh *et al*, 2000; Rosenwald *et al*, 2002; Benesova *et al*, 2013).

In summary, our study shows that standardised quantitative mass spectrometry-based proteomic analysis on FFPE tissues is a very promising and powerful tool for advanced tumour diagnostics and adds new information beyond nucleic acid-based methods. Importantly, once established, the complete workflow from a FFPE tumour sample down to the analysis can be performed within one week, which is comparable to the time required for next-generation sequencing workflows in pathology laboratories. The implementation of a protein quantification standard allows long-term comparability between individual samples, as well as between mass spectrometry platforms, with the perspective to leverage the enormous information embedded in FFPE tumour archives worldwide.

## Materials and Methods

### Tissue samples

Tissue samples were obtained from surgical resections at the Department of Thoracic Surgery and the Department of Otorhinolaryngology of the University Medical Centers Göttingen and Frankfurt a.M. None of the patients had received neoadjuvant chemotherapy. Approval for using the human patient material in this study was obtained from the Ethics Committee of the University Medical Center Göttingen (#1-2-08,9-12-10) and Frankfurt a.M. (#SKH-01-2016). Informed consent was obtained from all patients. All procedures were conducted in accordance with the Declaration of Helsinki and institutional, state and federal guidelines.

## Immunohistochemistry

Immunohistochemical reactions were performed as described previously (Mohr *et al*, 2017). Briefly, 2-μm tissue sections were incubated in EnVision Flex Target Retrieval Solution, pH high or low (Dako) followed by incubation of primary antibodies against CK5/6 (Dako, prediluted, high), CK7 (Dako, prediluted, high), p63 (Bio-Genex, prediluted, high), TTF-1 (Dako, prediluted, high), CD34 (Dako, prediluted, low), CK19 (Dako, prediluted, high), FTL (Atlas antibodies, 1:1,000, high), FAM83H (1:500, low), LGALS7 (Abcam, 1:20,000, high) or HMGCS1 (Atlas antibodies, 1:200, low) at room temperature for 20 min. Polymeric secondary antibodies coupled to HRPO peroxidase (EnVision Flex$^+$, Dako) and DAB (Dako) were applied to visualise the sites of immunoprecipitations. Tissue samples were analysed by light microscopy after counterstaining with Meyer's haematoxylin. Microvessel density was evaluated by staining of CD34 and counting of microvessels in five 10-fold magnification pictures. Tissue samples for immunohistochemical staining of CK19, FTL, FAM83H, LGALS7 and HMGCS1 were assembled in tissue microarrays prior to immunostaining, and all tissue samples were evaluated considering staining using a four-stage staining score multiplied with the percentage of positive stained tumour cells. The intensity of staining was evaluated on a graded scale (0 = negative; 1 = weakly positive; 2 = intermediately positive; 3 = strongly positive).

## TCGA data analysis

Mutational profile data for a cohort of 178 lung squamous carcinoma patients (LUSC, run date 20160128) and 279 head-and-neck squamous cell carcinoma patients (HNSC, run date 20160128) from The Cancer Genome Atlas (TCGA) were retrieved using R version 3.4.3 and the Bioconductor package RTCGAToolbox, version 2.6.0 (Samur, 2014). Mutation rates were computed with the getMutationRate function in RTCGAToolbox.

## HPV analysis

Head-and-neck squamous cell carcinoma samples were tested for 16 HPV types (HPV 6, 11, 16, 18, 31, 33, 35, 39, 45, 51, 52, 56, 58, 59, 66 and 68) using multiplex-based fluorescence PCR (F-HPV typing™, Genomed Diagnostics AG, *Altendorf,* Switzerland) and a subsequent fragment analysis using the ABI 3500 Genetic Analyzer (Life Technologies, Carlsbad, CA, USA). The data were analysed with the GeneMapper 5 Software (Life Technologies, CA, USA).

## Targeted next-generation sequencing

Targeted next-generation sequencing (NGS) was performed for genetic characterisation of all tumour samples. First, tumour areas were identified by a pathologist, and DNA was extracted from these areas by macrodissection followed by proteinase K treatment and automated extraction using the Maxwell 16 Research System (Promega, Madison, USA) or the InnuPureC16 System (Analytik-jena, Jena, Germany) following the manufacturers' protocols. The DNA content was measured using a real-time qPCR-based method.

The custom-made lung cancer panel consisted of 205 amplicons for the detection of mutations in 17 lung cancer-related genes

including *ARAF* exon 7; *BRAF* exons 11, 15; *CTNNB1* exon 3; *DDR2* exons 3–18; *EGFR* exons 18–21; *FGFR2* exons 8, 9, 10, 12, 17, 20; *FGFR3* exons 7, 10, 15; *HER2* exon 19, 20; *KEAP1* exons 2–6; *KRAS* exons 2–4; *MAP2K1* exon 2; *MET* exon 14, 16–19 and intron 14, 15; *NFE2L2* exon 2; *NRAS* exons 2–4; *PIK3CA* exons 9, 20; *PTEN* exons 1–8, and *TP53* exons 5–8. Isolated DNA (10 ng each) was amplified with four customised GeneRead Primer Pools (Qiagen). PCR products from the same patient were pooled after treatment with FuPa reagent. Following purification with Agencourt AMPure XP (Beckman Coulter, Brea, CA, USA), PCR products were incubated with NEXTflex™ DNA Adenylation Mix (Bioo Scientific Corp., Austin, TX, USA). NEXTflex™ DNA Barcodes were used as adapters (Bioo Scientific Corp.). After the bead size selection, NEXTflex™ PCR Master Mix (Bioo Scientific Corp.) was used for the final PCR amplification. Library products were quantified with a Qubit® 2.0 Fluorometer (Qubit® ds DNA HS Kit, Life Technologies™), diluted and pooled in equal amounts. 6–8 pM were spiked with 5% PhiX DNA (Illumina®, San Diego, CA, USA) and sequenced with the MiSeq™ reagent Kit V2 (300-cycles, Illumina®). Data were exported as FASTQ files. FASTQ files were aligned against reference NCBI build 37 (hg19) and annotated with a modified version of software previously described (Peifer *et al*, 2012). Resulting BAM files were visualised using the Integrative Genomics Viewer (IGV, Cambridge, USA). Called variants were then imported into a FileMaker (File-Maker GmbH, Germany) database for further analysis, annotation and reporting. A 5% cut-off for variant calls was used, and results were only interpreted if the coverage was >200×.

## Sample preparation for proteomic analysis

Tumour areas containing at least 80% tumour cells were marked by a pathologist on an H&E-stained slide, and corresponding areas of five sequential unstained 10 μm thick slides were isolated by macrodissection. Tumour samples were deparaffinised by incubations in xylene and absolute ethanol, each twice for 5 min at room temperature. Afterwards, samples were vacuum-dried and resuspended in lysis buffer containing 100 mM ABC, pH 8.0, 4% SDS, 0.2% DCA and 50 mM TCEP. Samples were incubated for 60 min at 90°C, sonicated for 3 min and centrifuged for 15 min at 20,000 $g$ at room temperature.

The Super-SILAC mix was composed of the cell lines NCI-H2228, HCC15, HCC44 and NCI-H1339. All cell lines were purchased from American Type Culture Collection (ATCC) and were metabolically labelled for more than 10 cell cycles with $^{13}C_6$-arginine (Arg + 6) and $D_4$-Lysine (Lys + 4) (Cambridge Isotope Laboratories) by culturing them in RPMI medium in which lysine and arginine were replaced by Arg + 6 and Lys + 4 and supplemented with 10% dialysed serum and antibiotics. For cell lysis, cells were washed with PBS and lysed with the same lysis buffer as described above. Lysates were incubated at 90°C for 15 min and then sonicated for 3 min.

Protein concentrations of the lysates were determined using the 660 nm and IDCR Kit (Thermo Fisher) following the manufacturer's instructions. Equal protein amounts of each clarified tissue lysate and Super-SILAC quantification standard were mixed for further filter-aided sample preparation (FASP) as described before (Wisniewski *et al*, 2009; Erde *et al*, 2014; Bohnenberger *et al*, 2015). Briefly, the mix of tissue lysate and quantification standard

**The paper explained**

**Problem**
Head-and-neck squamous cell carcinoma (HNSCC) is a frequently occurring disease, and long-term survival of patients with resectable HNSCC is often limited by lung metastasis. Because tobacco-smoking is a main risk factor for the development of HNSCC, HNSCC patients are also at high risk of developing lung cancer. In the context of HNSCC, differentiation between lung metastasis and secondary squamous cell lung cancer has important prognostic and therapeutic implications. However, reliable biomarkers for differentiation between these tumour types are lacking because of their shared morphologies and genomic features.

**Results**
Mass spectrometry-based protein expression profiling of HNSCC and squamous cell lung cancer samples revealed proteomic differences between the two tumour types. Using supervised machine learning, we identified a proteomic biomarker signature for differentiation of HNSCC-derived lung metastasis and primary lung tumours. This signature was validated in a cohort of lung tumours derived from patients with prior HNSCC.

**Impact**
This study demonstrates the feasibility and the added value of quantitative proteomics in advanced tumour diagnostics as it provides (i) a diagnostic molecular biomarker signature for differentiating between HNSCC-derived lung metastasis and metachronous squamous cell lung cancer in patients with prior HNSCC, and (ii) a proteomic resource for these tumour types.

was combined with 200 μl of 8 M urea and 0.2% deoxycholic acid (DCA) in 100 mM ABC, pH 8.0 and loaded onto a 30-kDa micron filter (Millipore). Subsequently, the filter was washed four times with the same buffer for 10 min at 14,000 $g$ at room temperature to rid the sample of SDS. Subsequently, protein samples were alkylated with 50 mM IAA in 100 mM ABC, pH 8.0, for 60 min, and urea was washed out three times with 0.2% DCA in 50 mM ammonium bicarbonate (ABC), pH 8.0, by centrifugation for 10 min at 14,000 $g$ at room temperature. Samples were digested overnight at 37°C with 1 μg of sequencing-grade trypsin (Promega), and peptides were collected by resuspending with 50 mM ABC, pH 8.0.

To remove deoxycholic acid and Tween-20 from the sample, the resulting peptide-containing solution was mixed with 200 μl ethyl acetate and 2.5 μl TFA. Thereafter, 200 μl ethyl acetate was added, samples were sonicated for 10 s and then centrifuged at 16,000 $g$ for 10 min at room temperature, and finally, the upper organic layer was removed. Samples were vacuum-dried and washed three times with 50% methanol. Afterwards, the peptides were desalted with ZipTips (Merck Millipore) or fractionated with Pierce™ High pH Reversed-Phase Peptide Fractionation Kit (Thermo Scientific) according to the manufacturer's instructions, and finally, resulting peptides were again vacuum-dried.

## Mass Spectrometry

The peptides were resuspended in sample loading buffer (2% acetonitrile and 0.05% trifluoroacetic acid) and were separated by an UltiMate 3000 RSLCnano HPLC system (Thermo Fisher Scientific) coupled online to either an Orbitrap Fusion or a Q Exactive HF. First, the peptides were desalted on a reverse-phase C18 pre-column (Dionex 5 mm long, 0.3 mm inner diameter) for 3 min. After 3 min, the pre-column was switched online with the analytical column (30 cm long, 75 μm inner diameter) packed in-house with ReproSil-Pur C18 AQ 1.9 μm reversed-phase resin (Dr. Maisch GmbH). Solvent A consisted of 0.1% formic acid in water, and solvent B consisted of 80% acetonitrile and 0.1% formic acid in water. The peptides were eluted from buffer B (5–42% gradient) at a flow rate of 300 nl/min over 166 min. The temperature of the pre-column and the column was set to 50°C during the chromatography. MS data were acquired on an Orbitrap Fusion in data-dependent Top speed mode, where numerous precursors within the m/z range of 350–1,500 Th and charge state of 2–6 were selected from a survey MS1 scan for MS2 fragmentation with an isolation window of 1.6 Th and a maximum cycle time of 3 s. MS1 scans were acquired at a resolution of 120,000 and AGC target of 4E5. Selected precursors underwent HCD fragmentation with a normalised collision energy (NCE) of 35. MS2 scans were acquired in the ion trap in rapid-scan mode with maximum ion injection (IT) time of 250 ms and AGC target of 3E3. All scan events were automatically parallelised. Dynamic exclusion (DE) was set to 60 s. For the measurement on the Q Exactive HF, the mass spectrometer was operated in Top 30 data-dependent mode, where the most intense 30 precursors within the m/z range of 350–1,500 were selected for MS2 fragmentation with an NCE of 28. MS2 spectra were acquired in the Orbitrap with a resolution of 15,000 and maximum IT of 60 ms. AGC for MS1 and MS2 scans were 1E6 and 1E5, respectively. DE was 45 s. Other settings were the same as the ones on the Orbitrap Fusion.

## MS data processing

MS/MS spectra were searched against a UniProtKB/Swiss-Prot human database containing 134,921 protein entries (downloaded in July 2014) supplemented with 245 frequently observed contaminants collated with the Andromeda search engine (Cox *et al*, 2011). Precursor and fragment-ion mass tolerances were set to 6 and 20 ppm, respectively, after initial recalibration. Protein N-terminal acetylation and methionine oxidation were allowed as variable modifications. Cysteine carbamidomethylation was defined as a fixed modification. Minimum peptide length was set to seven amino acids, with a maximum of two missed cleavages. The false discovery rate (FDR) was set to 1% at both the peptide and the protein level using a forward-and-reverse concatenated decoy database approach.

For SILAC quantification, multiplicity was set to two for double SILAC (Lys + 0/Arg + 0, Lys + 8/Arg + 10) labelling. At least two ratio counts were required for peptide quantification. The "match between runs" option of MaxQuant was disabled, and the "re-quantify" option was enabled.

## Intersection of proteomic and CRISPR/Cas9 loss-of-function screen data

In order to identify whether the 518 differentially expressed proteins in HNSCC and SQCLC showed evidence of functional dependency in these tumour types, we interrogated publicly available data

(https://depmap.org/portal/dataset/download/Avana/portal-Avana-2018-04-11.csv) from a genome-scale CRISPR-Cas9 functional screen using the Avana 1.0 library in 391 cancer cell lines that included 12 HNSCC (BHY, BICR22, BICR56, BICR6, CAL27, CAL33, DETROIT562, FADU, HSC3, PECAPJ34CLONEC12, UPCISCC152, YD38) and 10 SQCLC (HARA, HCC15, HCC95, KNS62, LK2, LUDLU1, NCIH2170, NCIH520, RERFLCAI, SKMES1) cell lines (Doench *et al*, 2016; Meyers *et al*, 2017). Not all differentially expressed proteins identified in our study were included in the CRISPR-Cas9 screening results, and therefore, this analysis was limited to those that overlapped (*n* = 471/518 differentially expressed proteins, 28/30 HNCSC highly expressed proteins, 73/79 SQCLC highly expressed proteins). We performed a two-class comparison between genetic dependencies in HNSCC and SQCLC with a difference of means two-sided *t*-test and intersected these data with the proteins that were identified as differentially expressed between HNSCC and SQCLC.

### Statistical data analysis

Data analysis was performed in R version 3.4.1 (R-Core-Team, 2017) and Bioconductor version 3.2 (Huber *et al*, 2015). Due to the agnostic approach pursued, no prior sample size computation was performed; all available samples were included in the analysis. Proteomic data were processed as follows: Proteins with more than 2/3 missing values were excluded from further analysis. For the remaining 2,586 proteins, missing values were imputed using *k*-nearest-neighbour imputation with $k = 10$. Univariate analysis using the empirical Bayes approach was performed with the Bioconductor package *limma* (Ritchie *et al*, 2015), using the lmFit and eBayes functions. *P* values were corrected using the false discovery rate. For tests involving comparisons of means, data were assessed for normality and similarity of variances between groups, and a *t*-test (with correction for unequal variances when appropriate) or a Wilcoxon test was chosen accordingly. Multivariate analysis was performed using the R package *caret* (Kuhn, 2008). Support vector machine training was performed using bootstrapping with 25 repetitions, with parameter C = 1 in the SVM and a linear kernel. Feature selection was done using recursive feature elimination. Survival analysis was performed using the Kaplan–Meier estimator and the log-rank test. In all analyses, the investigator was blinded for tumour types in the validation cohort and for patient outcome for the classification of tumours of unknown origin.

## Data availability

All of the raw files and MaxQuant search results have been deposited in the ProteomeXchange Consortium (http://proteomecentral.proteomexchange.org/cgi/GetDataset) via the PRIDE partner repository (Vizcaino *et al*, 2014) with the dataset identifier PXD007705.

**Expanded View** for this article is available online.

## Acknowledgements

We thank Jennifer Appelhans, Pamela Nissen, Monika Raabe, Annika Kühn, Silvia Münch and Martine Pape for their technical support. H.B., L.H.D and K.R-J. are supported by the Else-Kröner-Fresenius Foundation. H.B. and T.O. are supported by the Deutsche Krebshilfe Foundation (grant: 70112551). L.K. received funding from the Deutsche Krebshilfe Foundation (grant: 70112014). Graphical artwork for Fig 2A and the Synopsis figure was modified from versions provided by Servier Medical Art, licensed under a Creative Common Attribution 3.0 Generic License.

## Author contributions

HB and TO conceived and supervised the project. LK performed bioinformatics and statistical analyses. FC and NVD performed bioinformatic analyses. HB, TO, HU, DY, NVD, KS, UP, K-TP, SM-B, RB, RW, SY, JB, SK, BH, A-ML, JH, MS, TB, KR-J, KS, CL, CH and H-US performed and/or supervised experiments. AE, LHD, LB, PB, CW, SS, FB, JK, MH, JL, HW, BCD, MC and CB contributed clinical samples and/or patient characteristics. MS, HS, MC and PS provided discussion. HB, LK, PS and TO wrote the manuscript with final approval of all authors.

## Conflict of interest

The authors declare that they have no conflict of interest.

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
