## [Review Process File · EMBO Molecular Medicine]

Comparative proteomics reveals a diagnostic signature for pulmonary head-and-neck cancer metastasis

Hanibal Bohnenberger, Lars Kaderali, Philipp Ströbel, Diego Yepes, Uwe Plessmann, Neekesh V. Dharia, Sha Yao, Carina Heydt, Sabine Merkelbach-Bruse, Alexander Emmert, Jonatan Hoffmann, Julius Bodemeyer, Kirsten Reuter-Jessen, Anna-Maria Lois, Leif Hendrik Dröge, Philipp Baumeister, Christoph Walz, Lorenz Biggemann, Roland Walter, Björn Häupl, Federico Comoglio, Kuan-Ting Pan, Sebastian Scheich, Christof Lenz, Stefan Küffer, Felix Bremmer, Julia Kitz, Maren Sitte, Tim Reißbarth, Marc Hinterthaler, Martin Sebastian, Joachim Lotz, Hans-Ulrich Schildhaus, Hendrik Wolff, Bernhard C. Danner, Christian Brandts, Reinhard Büttner, Martin Canis, Kimberly Stegmaier, Hubert Serve, Henning Urlaub, Thomas Oellerich

Review timeline:

Submission date:	19 September 2017
Editorial Decision:	15 November 2017
Revision received:	30 April 2018
Editorial Decision:	25 June 2018
Revision received:	09 July 2018
Accepted:	10 July 2018

Editor: Céline Carret

Transaction Report:

1st Editorial Decision

15 November 2017

Thank you for the submission of your manuscript to EMBO Molecular Medicine. We have now heard back from the three referees whom we asked to evaluate your manuscript.

As you will see from the reports below, the referees find the topic of your study of potential interest. However they all agree that the paper cannot be published in this form: additional figures/table are suggested, more state of the art analyses are requested, along with better explanations/clarifications of the experimental set up and analysis strategy, and (importantly for our scope) biological / functional insights are needed. We also would appreciate a discussion on how these data could be useful in the clinic.

Overall it is clear that publication of the manuscript cannot be considered at this stage. I also note that addressing the reviewers concerns in full will be necessary for further considering the manuscript in our journal and this appears to require a lot of additional work and experimentation, maybe of a nature that does not appeal to you. I am unsure whether you will be able or willing to address those and return a revised manuscript within the 3 months deadline. On the other hand, given the potential interest of the findings, I would be willing to consider a revised manuscript with the understanding that the referee concerns must be fully addressed and that acceptance of the manuscript would entail a second round of review. I should remind you that EMBO Molecular Medicine encourages a single round of revision only and that, therefore, acceptance or rejection of the manuscript will depend on the completeness of your responses included in the next, final version of the manuscript. For this reason, and to save you from any frustrations in the end I would strongly advise against returning an incomplete revision and would also understand your decision if you chose to rather seek rapid publication elsewhere at this stage.

Should you find that the requested revisions are not feasible within the constraints outlined here and choose, therefore, to submit your paper elsewhere, we would welcome a message to this effect.

I look forward to receiving your revised manuscript.

***** Reviewer's comments *****

Referee #1 (Remarks for Author):

Summary

Bohnenberger and colleagues performed a comparative proteomics analysis of patient-derived head-and-neck squamous-cell carcinoma (HNSCC) and squamous cell carcinomas of the lung (SQCLC) tissues with the aim to establish a diagnostic biomarker signature to distinguish these two tumor types. Following immunohistochemical and mutational characterization, the authors analyzed the protein extracts of 74 microdissected tumor samples by super-SILAC-based quantitative mass spectrometry. Using extensive statistical analysis, Bohnenberger et al. unveiled more than 500 tumor-type specific, differentially regulated proteins within a total of approximately 6000 quantified proteins. Next, the authors fed this proteomic data set into a machine-learning pipeline. Using only 1100 proteins of this dataset, the predictor enabled discrimination between SQCLC and HNSCC with a high degree of accuracy, sensitivity and specificity. Finally, the authors examined the predictive power of their diagnostic proteomic signature by employing another set of 30 squamous-cell lung tumors which cannot be classified as HNSCC or SQCLC based on clinical classification. Remarkably, the proteomics-based biomarker signature allowed differentiation between the two tumor types. Together, this study is very well rationalized, controlled and conducted and impressively shows how comprehensive proteomics analysis can be capitalized for the development of diagnostic-relevant biomarkers. I only have a few minor points.

Minor points

- 1) Given the mutation profile of HNSCC or SQCLC revealed by the sequencing analysis, the authors should map protein abundance changes on those pathways which found to be most affected by mutations (i.e. PI3-kinase and Ras pathways, receptor tyrosine kinases, TP53 and the NFE2L2/KEAP1 pathway).
- 2) It would be very informative to show the ten most differentially regulated proteins in both tumor type and show their variance across samples within each tumor type.

Referee #2 (Remarks for Author):

This manuscript describes the generation of a proteomic signature that discriminates lung HNSCC metastasis from primary SQCLC that arise in patients that previously developed HNSCC. My main concern is that the validation experiment is done with most of the patients used in the initial generation of the proteomic signatures. Lung metastases share genomic alterations and expression patterns with their primary tumors. Therefore, the proteomic signatures generated using primary tumors as a training set are expected to be conserved when applied to their own metastasis, to a certain degree. It is unclear whether the signatures generated from this cohort would translate to a totally independent cohort.

Also, it is not clear the clinical relevance of the proteomic signatures from a practical point of view; they should at least discuss how they envision that these findings can be translated to the clinic.

The difference in survival is an indirect confirmation of the tumor identity.

I would also ask the authors to mention how many patients in the validating cohort were assigned to HNSCC metastasis and primary SQCLC by the proteomic signatures. A table comparing the tumor type assignment for each tumor according to the clinical score and the proteomic signature would be helpful, adding the p53 status of those tumors/metastases.

Referee #3 (Remarks for Author):

In this manuscript, the authors report a panel of proteins that enables the differentiation of head-and-neck squamous-cell carcinomas (HNSCCs) that metastasized into the lung from HNSCCs that developed squamous cell lung carcinomas (SQCLC). The differentiation between the two tumor types is relevant, as therapeutic procedures are quite different and associated with different outcome. To this end, the authors measured the proteomes of 44 and 30 FFPE SQCLC and HNSCC tissues, respectively. Approximately 2,250 proteins were quantified per sample. They report 518 proteins with significant different levels of expression in HNSCC and SQCLC. Using all quantified proteins as features and the 74 tissues as samples, they trained a linear support vector machine (SVM) to discriminate HNSCC from SQCLC. The authors found a signature of 1,100 proteins that enables differentiation of HNSCC from SQCLC. A major positive point of the manuscript is that the authors validated this proteomic signature in an independent cohort (only 30 samples though).

This is a generally a well done study, however, the methods are not quite state of the art any longer, and consequently the depth of proteomics coverage is rather shallow compared to what can be done today. Combined with the lack of biologically meaningful differences between the subtypes, this is a weakness of the manuscript.

Major concerns

- The authors did not describe any biological function of the most significantly expressed proteins in HNSCC vs. SQCLC. Do these proteins make biologically sense to enable differentiation between HNSCC and SQCLC?

- Why did the authors use empirical Bayes method to compare the HNSCC and SQCLC groups and not a regular t-test (Figure 1F)?

- Ferritin light chain (FTL) was found as the most significant protein enabling differentiation of HNSCC vs. SQCLC (Supplementary Table 5). FTL is highly abundant in plasma, is this only an artefact of one tumor being more vascularized than the other one?

- The signature, comprising 1,100 proteins seems quite a lot. It represents almost half of all quantified proteins. Signatures of less than 20 proteins are more typical in these kinds of support vector machine (SVM) classifications:

- The authors neither describe receiver operating characteristics (ROC) curves, nor how the predictors are performing based on area under the curve (AUC).

- Why did the authors use Bootstrapped accuracy and not cross validation of the data?

- What is the standard deviation of the bootstrapped accuracies? Please add them in Figure 2A.

- How is this signature improving the currently clinical classification of HNSCC vs. SQCLC in the first cohort?

- Authors next applied this proteomic signature in a second cohort (30 samples). It would be desirable to increase the size of the second cohort. Why did the authors only include 8 samples that have been classified as primary lung cancer and not half of the samples?

- How is the classification accuracy performing for the new cohort?

- It is not clear to me how the authors came up with survival time. Why are the first set of samples not classified according to the survival time?

- The authors report that genomic and transcriptomic studies have not been able to differentiate the 2 types. The authors only show mutation data for PI3K, RTK, TP53, KEAP, HPV, and RAS. Have other genes been analyzed?

- MS measurements: 166 min gradients using super SILAC spike in for quantification. Where all samples measured on an Orbitrap Fusion and/or Q Exactive HF?

- Can you comment on the technical reproducibility of the proteomics measurements?

- Is there access to the raw data on PRIDE?

Minor concerns

- A figure describing the experimental workflow would be desirable, in particular for the computational workflow.
- The authors should increase the current proteomic depth using fractionation (e.g. high pH reversed phase) of the samples and see whether the current proteomic signature can still be applied in the context of deeper proteomes.
- Be consistent throughout the manuscript and use the 'metHNSCC' abbreviation.
- Manual microdissection should be replaced with macro dissection.
- Regarding sample preparation, I would expect important losses using FASP after manual microdissection. Can you comment on the sample loss?

1st Revision - authors' response

30 April 2018

Reviewer #1:

Bohnenberger and colleagues performed a comparative proteomics analysis of patient-derived head-and-neck squamous-cell carcinoma (HNSCC) and squamous cell carcinomas of the lung (SQCLC) tissues with the aim to establish a diagnostic biomarker signature to distinguish these two tumor types. Following immunohistochemical and mutational characterization, the authors analyzed the protein extracts of 74 microdissected tumor samples by super-SILAC-based quantitative mass spectrometry. Using extensive statistical analysis, Bohnenberger et al. unveiled more than 500 tumor-type specific, differentially regulated proteins within a total of approximately 6000 quantified proteins. Next, the authors fed this proteomic data set into a machine-learning pipeline. Using only 1100 proteins of this dataset, the predictor enabled discrimination between SQCLC and HNSCC with a high degree of accuracy, sensitivity and specificity. Finally, the authors examined the predictive power of their diagnostic proteomic signature by employing another set of 30 squamous-cell lung tumors which cannot be classified as HNSCC or SQCLC based on clinical classification. Remarkably, the proteomics-based biomarker signature allowed differentiation between the two tumor types. Together, this study is very well rationalized, controlled and conducted and impressively shows how comprehensive proteomics analysis can be capitalized for the development of diagnostic-relevant biomarkers. I only have a few minor points.

We are naturally very pleased to read the reviewer's positive comments on our work.

- 1. Given the mutation profile of HNSCC or SQCLC revealed by the sequencing analysis, the authors should map protein abundance changes on those pathways which found to be most affected by mutations (i.e. PI3-kinase and Ras pathways, receptor tyrosine kinases, TP53 and the NFE2L2/KEAP1 pathway).**

We have addressed this point by extracting proteins related to these pathways from the KEGG database (these were 258 proteins related to PI3K signaling, 62 to TP53, and 24 to the NFE2L2/KEAP1 pathway) and then analysing their protein expression profiles in the SQCLC and HNSCC samples of the training and validation cohorts. Interestingly, we did not observe differential expression of any of these proteins between the HNSCC and SQCLC samples, and neither did we observe differential expression between the samples with and without mutations in these pathways. This does not exclude the possibility that there are differences in pathway activity, for instance mediated by phosphorylation, which we cannot detect by quantitative protein expression profiling. We have mentioned these observations in the results section.

- 2. It would be very informative to show the ten most differentially regulated proteins in both tumor type and show their variance across samples within each tumor type.**

We have included these data in supplementary figure 3 of the revised manuscript.

Reviewer #2:

This manuscript describes the generation of a proteomic signature that discriminates lung HNSCC metastasis from primary SQCLC that arise in patients that previously developed HNSCC.

The reviewer has indeed identified a very important aspect, which we have now addressed by validating our diagnostic proteomic signature in independent patient cohorts as described in detail below.

- 1. My main concern is that the validation experiment is done with most of the patients used in the initial generation of the proteomic signatures. Lung metastases share genomic alterations and expression patterns with their primary tumors. Therefore, the proteomic signatures generated using primary tumors as a training set are expected to be conserved when applied to their own metastasis, to a certain degree. It is unclear whether the signatures generated from this cohort would translate to a totally independent cohort.**

This point is well taken. As suggested, we have now validated our diagnostic proteomic signature in independent patient cohorts.

Because the definite tumour-of-origin for the lung tumours of patients with prior HNSCC cannot be determined by any diagnostic tool, we decided to validate our diagnostic signature first in an independent cohort of confirmed primary lung (SQCLC) and head-and-neck cancer (HNSCC) cases. Therefore, we have now analysed additional 19 independent cases of each tumour type using our quantitative proteomic pipeline and have subsequently applied our diagnostic signature to classify these tumours as either HNSCC and SQCLC.

Similarly to the initial analysis, the diagnostic accuracy in this independent validation cohort was 86.8%, the sensitivity 85%, and the specificity 88.9% (see below). We have included this new result in Fig. 4E.

	True HNSCC	True SQCLC
Predicted HP	16	2
Predicted PP	3	17
Diagnostic accuracy: 86,8%		
Sensitivity: 85%		
Specificity : 88,9%		

In addition, we have now analysed 21 additional lung tumour samples of unknown origin derived from patients with prior HNSCC to expand our second validation cohort by including additional cases that were independent of the training cohort. Again, we found significant prognostic separation (as a surrogate marker for correct classification) by classifying the expanded cohort of tumours of unknown origin using our diagnostic signature (Fig. 4G). Notably, we also observed a similar prognostic separation when we excluded all cases that were related to the training cohort (Fig. 4H). This result provides further evidence for the validity of our approach and for the diagnostic signature that we identified.

- 2. Also, it is not clear the clinical relevance of the proteomic signatures from a practical point of view; they should at least discuss how they envision that these findings can be translated to the clinic.**

As suggested, we have included a paragraph addressing this point in the discussion.

- 3. I would also ask the authors to mention how many patients in the validating cohort were assigned to HNSCC metastasis and primary SQCLC by the proteomic signatures. A table comparing the tumor type assignment for each tumor according to the clinical score and the proteomic signature would be helpful, adding the p53 status of those tumors/metastases.**

Of the 51 cases with lung tumours of unknown origin, we classified 42 as primary lung tumours and 9 as HNSCC metastasis using our diagnostic signature. We have now included this information, including the p53 status, in supplementary figure 4C.

Reviewer #3:

In this manuscript, the authors report a panel of proteins that enables the differentiation of head-and-neck squamous-cell carcinomas (HNSCCs) that metastasized into the lung from HNSCCs that developed squamous cell lung carcinomas (SQCLC). The differentiation between the two tumor types is relevant, as therapeutic procedures are quite different and associated with different outcome.

To this end, the authors measured the proteomes of 44 and 30 FFPE SQCLC and HNSCC tissues, respectively. Approximately 2,250 proteins were quantified per sample. They report 518 proteins with significant different levels of expression in HNSCC and SQCLC. Using all quantified proteins as features and the 74 tissues as samples, they trained a linear support vector machine (SVM) to discriminate HNSCC from SQCLC. The authors found a signature of 1,100 proteins that enables differentiation of HNSCC from SQCLC. A major positive point of the manuscript is that the authors validated this proteomic signature in an independent cohort (only 30 samples though). This is a generally a well done study, however, the methods are not quite state of the art any longer, and consequently the depth of proteomics coverage is rather shallow compared to what can be done today. Combined with the lack of biologically meaningful differences between the subtypes, this is a weakness of the manuscript.

We appreciate this referee's detailed review of our manuscript.

- 1. The authors did not describe any biological function of the most significantly expressed proteins in HNSCC vs. SQCLC. Do these proteins make biologically sense to enable differentiation between HNSCC and SQCLC?**

To gain insights into the functional relevance of the proteins that we found to be differentially expressed in SQCLC and HNSCC, we have integrated our proteomic results with recently published functional genomic data, i.e. data from CRISPR/Cas9-based drop-out screens (<https://depmap.org/portal/dataset/download/Avana/portal-Avana-2018-04-11.csv>) (Meyers RM, et al. Computational correction of copy number effect improves specificity of CRISPR-Cas9 essentiality screens in cancer cells. *Nat Genet.* 2017 Dec;49(12):1779-1784. Doench JG, et al. Optimized sgRNA design to maximize activity and minimize off-target effects of CRISPR-Cas9. *Nat Biotechnol.* 2016 Feb;34(2):184-191). In detail, we analyzed if any of the 518 differentially expressed proteins were identified as genetic dependencies (which means regulators of proliferation and/or cell survival) in cell lines of either of the two tumour types (12 HNSCC and 10 SQCLC cell lines). Interestingly, this analysis revealed, that some of the differentially expressed proteins are preferential genetic dependencies in HNSCC and SQCLC (Fig. 3B, Supplementary Fig. 3B, Supplementary Table 3). This result highlights that some of the differentially expressed marker proteins are relevant for regulating cell survival and/or proliferation in the respective tumour types.

- 2. Why did the authors use empirical Bayes method to compare the HNSCC and SQCLC groups and not a regular t-test (Figure 1F)?**

The traditional t-test is known to have problems with low expression values when the sample size is small (Smyth, *Stat Appl Genet Mol Biol* 2004). The empirical Bayes approach used is a moderated version of the t-test that does not suffer from this disadvantage. Essentially, the estimated sample variances are shrunk towards a pooled estimate. This results in a far stabler inference, in particular if the number of samples is small.

- 3. Ferritin light chain (FTL) was found as the most significant protein enabling differentiation of HNSCC vs. SQCLC (Supplementary Table 5). FTL is highly abundant in plasma, is this only an artefact of one tumor being more vascularized than the other one?**

We have addressed this point by analysing FTL expression in 212 SQCLC and 343 HNSCC samples (see Fig. 3C). Both expression of FTL in tumour cells and significantly higher expression levels in SQCLC than in HNSCC cells could be confirmed by expert pathology review. In addition, we have quantified the microvessel density in most tumours of the training cohort and found no significant differences between SQCLC and HNSCC (Supplementary Fig. 2C). Taken together, these results provide clear evidence that the FTL derived from the tumour cells.

4. The signature, comprising 1,100 proteins seems quite a lot. It represents almost half of all quantified proteins. Signatures of less than 20 proteins are more typical in these kinds of support vector machine (SVM) classifications.

To follow up on this comment, we have used recursive feature elimination to address the effect of reduced signatures. Our results show that below 1,100 proteins the classification performance starts to drop rapidly (Fig. 4C). This finding indicates that a reliable separation of such closely related tumour types is difficult and requires a large proteomic signature. Similar observations have been made for other closely related tumour types. For example, activated B-cell-like and germinal centre B-cell-like diffuse large-B-cell lymphomas can only be separated reliably by complex gene expression signatures, while less complex diagnostic markers appear to be unreliable (*Alizadeh AA, et al (2000) Distinct types of diffuse large B-cell lymphoma identified by gene expression profiling. Nature 403: 503-511 ; Rosenwald A, et al (2002) The use of molecular profiling to predict survival after chemotherapy for diffuse large-B-cell lymphoma. N Engl J Med 346: 1937-1947 ; Benesova K et al (2013) The Hans algorithm failed to predict outcome in patients with diffuse large B-cell lymphoma treated with rituximab. Neoplasma 60: 68-73*). We have discussed this aspect in the discussion section of the revised manuscript.

5. The authors neither describe receiver operating characteristics (ROC) curves, nor how the predictors are performing based on area under the curve (AUC).

Support vector machines (SVM) perform a binary classification, assigning each probe to exactly one of two conditions (here, HNSCC or SQCLC). If these class predictions are compared with the true classes, exactly one value for sensitivity and specificity arises, corresponding to a single point in an ROC curve.

Plotting ROC curves would require a spread of different values for sensitivity and specificity, as would be available if different thresholds on a classification “score” were chosen for class assignments. If these points are then connected, ROC curves arise. AUC in turn is computed as the area under the ROC curve. However, as SVMs provide no score, this is in our case not possible.

Sometimes, ROC curves are computed for SVM to estimate model parameters of the SVM. Different points in the ROC curve then correspond to different SVM parameters. However, we optimised model parameters by using a grid search in bootstrapped training of the classifier; therefore, such ROC curves would not be useful here.

6. Why did the authors use Bootstrapped accuracy and not cross validation of the data?

Bootstrapping is an alternative approach to internal validation of a model, and is particularly useful if robustness of a predictor is of major interest. Robustness is particularly important when one is comparing different predictors; it assesses directly the variance of the model's performance. In comparison with crossvalidation, bootstrapping may result in slightly higher bias of the performance estimates. To control for this effect, we performed additional model validation on completely independent data sets.

7. What is the standard deviation of the bootstrapped accuracies? Please add them in Figure 2A.

We have now added the standard deviations to the former Fig. 2A (now Fig. 4B).

8. How is this signature improving the currently clinical classification of HNSCC vs. SQCLC in the first cohort?

The thrust of this query is not entirely clear to us. The first cohort, which consisted of 44 primary SQCLC and 30 primary HNSCC cases, was used as a training cohort to extract a protein expression signature that allows differentiation between HNSCC and SQCLC. To get reliable results we included in the training cohort only primary SQCLC and HNSCC cases that had been confirmed as such by expert pathology review. Clinically, differentiation between primary HNSCC and SQCLC is trivial, as these tumours grow in different tissues / anatomic regions. Hence, there is no diagnostic problem in differentiating between primary HNSCC and primary SQCLC. However, the differentiation between HNSCC metastasis in the lung and primary squamous cell lung cancers is challenging owing to the lack of biomarkers.

If the reviewer's question is rather referring to possible overfitting of the model to the training data, we may point out that we have now validated our diagnostic proteomic signature in independent patient cohorts. Because the definite tumour-of-origin for the lung tumours of patients with prior HNSCC cannot be determined by any diagnostic tool, we decided first of all to validate our diagnostic signature in an independent cohort of confirmed primary SQCLC and HNSCC cases. To this end we analysed additional 19 independent cases of each tumour type using our quantitative proteomic pipeline and subsequently applied our diagnostic signature to classify these tumours as either HNSCC and SQCLC.

Similarly to our former crossvalidation approach, the diagnostic accuracy in this independent validation cohort was 86.8%, the sensitivity 85%, and the specificity 88.9%. We have included this new result in Fig. 4E.

9. Authors next applied this proteomic signature in a second cohort (30 samples). It would be desirable to increase the size of the second cohort.

We agree with the reviewer that this is an important point. In addition to our validation efforts described above, we have now analysed 21 additional lung tumour samples of unknown origin derived from patients with prior HNSCC to expand our second validation cohort by including additional cases that were independent of the training cohort. Again, we found significant prognostic separation (as a surrogate marker for correct classification) by classifying the expanded cohort of tumours of unknown origin using our diagnostic signature. Notably, we also observed a similar prognostic separation when we excluded all cases that were related to the training cohort. This result provides further evidence for the validity of our approach and of the diagnostic signature that we identified.

10. Why did the authors only include 8 samples that have been classified as primary lung cancer and not half of the samples?

We have now analysed all of the lung tumour samples from patients with prior HNSCC to which we had access. These were in total 51 cases, of which 11 and 40 had originally been classified clinically as being derived from SQCLC and metHNSCC, respectively. The numbers are a result of the availability of the clinically characterized tumour samples.

**11. How is the classification accuracy performing for the new cohort?
and**

12. It is not clear to me how the authors came up with survival time. Why are the first set of samples not classified according to the survival time?

In the revised version of our manuscript we have now analysed 3 patient cohorts.

1. A training cohort (cohort 1; 44 primary SQCLC and 30 primary HNSCC cases)
2. An independent validation cohort (cohort 2; 19 primary SQCLC and 19 primary HNSCC cases)
3. A cohort of 51 lung tumours of unknown origin derived from patients with prior HNSCC (cohort 3). These tumours could have been either HNSCC metastases or primary localised lung cancers (SQCLC)

As described above (point 8 above), we have now validated our proteomic diagnostic signature in an independent validation cohort (n=38; cohort 2), which consisted of independent primary SQCLC and HNSCC cases. In this cohort our diagnostic signature achieved an accuracy of 86.8% (Fig. 4E). As the “true class” (SQCLC or HNSCC) is known for cohorts 1 and 2, predictive performance can be assessed directly by using common performance measures (sensitivity, specificity, accuracy etc.).

The diagnostic accuracy of our approach for classifying lung tumours of unknown origin derived from patients with prior HNSCC (cohort 3) as either SQCLC or HNSCC metastasis cannot be accurately assessed, because there is no diagnostic tool / gold standard that can distinguish between SQCLC and HNSCC metastasis. However, because a prognostic disadvantage would be expected for patients with metastatic HNSCC compared with localised primary SQCLC, we used overall survival as a surrogate marker for the diagnostic accuracy in this cohort. Interestingly, our proteomic signature achieved the expected prognostic separation (Fig. 4G–H), in contrast to the original classification, which was based on largely unvalidated clinical and/or imaging criteria (*Ichinose J, Shinozaki-Ushiku A, Nagayama K, Nitadori J, Anraku M, Fukayama M, Nakajima J, Takai D (2016) Immunohistochemical pattern analysis of squamous cell carcinoma: Lung primary and metastatic tumors of head and neck. Lung Cancer 100: 96-101*).

We hope that this additional information will clarify the matter.

13. The authors report that genomic and transcriptomic studies have not been able to differentiate the 2 types. The authors only show mutation data for PI3K, RTK, TP53, KEAP, HPV, and RAS. Have other genes been analyzed?

We have characterised our training cohort by a panel-sequencing approach that covered the 17 most recurrently mutated genes in SQCLC/HNSCC. To address the reviewer's question, we have now extracted the mutation patterns and frequencies of primary SQCLC and HNSCC from more comprehensive genomic studies already published (*Cancer Genome Atlas N (2015) Comprehensive genomic characterization of head and neck squamous cell carcinomas. Nature 517: 576-582; Cancer Genome Atlas Research N (2012) Comprehensive genomic characterization of squamous cell lung cancers. Nature 489: 519-525*). In accordance with our results, the mutation frequencies are strongly correlated between the two tumour types (Fig. 1C), supporting our statement that HNSCC and SQCLC mutation patterns generally largely overlap and are therefore not useful as biomarkers.

14. MS measurements: 166 min gradients using super SILAC spike in for quantification. Where all samples measured on an Orbitrap Fusion and/or Q Exactive HF?

All samples were analysed on an Orbitrap Fusion and/or Q exactive HF mass spectrometer. The standardised work-flow means that the results were comparable and reproducible, as shown for 6 randomly selected samples that were analysed on both mass spectrometers (Supplementary Fig. 1B and C).

15. Can you comment on the technical reproducibility of the proteomics measurements?

The proteomic measurements were found to be highly reproducible. Experiments were done in two replicates. Pearson correlation between these individual replicates ranged from 0.802 to 0.975 per sample, with a mean of 0.949 and a standard deviation of 0.0266. We have added the relevant details to the results section.

16. Is there access to the raw data on PRIDE?

We have submitted our raw and processed mass-spectrometry data to the PRIDE proteomics data repository (<http://www.ebi.ac.uk/pride/archive/>), to allow thorough evaluation of the results by both reviewers and readers (Project accession: PXD007705; Account details: reviewer14964@ebi.ac.uk; password: 83HbIhIC).

17. A figure describing the experimental workflow would be desirable, in particular for the computational workflow.

Figures 2A and 4A have been added to illustrate the proteomic and computational workflows.

18. The authors should increase the current proteomic depth using fractionation (e.g. high pH reversed phase) of the samples and see whether the current proteomic signature can still be applied in the context of deeper proteomes.

The major goal of our study was to identify a proteomic biomarker signature for differentiation between HNSCC metastasis in the lung on the one hand and primary SQCLC on the other. This is of clinical importance because HNSCC patients are at risk of developing both of these, and an accurate diagnosis is needed to guide the selection of therapeutic procedures (potentially curative treatment for SQCLC, palliative treatment of metHNSCC). Because of the primarily clinical/diagnostic scope of our study, we have decided to use a more time-efficient and less complex experimental workflow. In our experience this allows more accurate quantification, which we regard as very important in a diagnostic context; moreover, rapid MS acquisition and data analysis are better suited for the clinical setting. In the light of these considerations we decided to give priority to the more rapid and accurate work-flow.

As the SVM framework is based on quantitative computations, deeper coverage (as achieved through the more time-consuming fractionation procedure) might result in numerical changes of the measured values, that could potentially affect classification results. Hence, data would either need to be normalised so that ranges are comparable to the values used for training the SVM, or re-estimation of the SVM coefficients for the signature proteins would have to be performed to take account of the deeper coverage. Hence, a standardised assay would be needed for diagnostic purposes, comparable to the standardised next-generation sequencing-based assays that are used in routine diagnostics.

19. Be consistent throughout the manuscript and use the 'metHNSCC' abbreviation.

The text has been changed accordingly.

20. Manual microdissection should be replaced with macro dissection.

The text has been changed accordingly.

21. Regarding sample preparation, I would expect important losses using FASP after manual microdissection. Can you comment on the sample loss?

Manuel macrodissection is a well-established technique for genetic analysis of tumour tissue samples in routine pathological diagnostics. Areas of tumour cells are marked and purified under the microscope by an expert pathologist. After protein extraction, we were able to recover 60–70% of the input material after ‘filter-aided’ sample preparation and tryptic digestion.

2nd Editorial Decision

25 June 2018

Thank you for the submission of your revised manuscript to EMBO Molecular Medicine. We have now received the enclosed reports from the referee who was asked to re-assess it. As you will see this reviewer is now globally supportive and I am pleased to inform you that we will be able to accept your manuscript pending minor editorial amendments.

***** Reviewer's comments *****

Referee #1 (Remarks for Author):

The authors did a very good job in adequately addressing my concerns. They significantly extended the proteomics of patient samples to improve the machine learning approach, performed additional analyses of their data sets and revised their manuscript.

Point1 of referee1: I could not find any reference to this in the result section. Would be nice to include this analysis as supplemental figure.

I do not have any more objections to recommend this manuscript for publication.

Corresponding Author Name: Thomas Oellerich

Journal Submitted to: Embo Molecular medicine

Manuscript Number: EMM-2017-08428